# Factors Associated with Birth-Related Post-Traumatic Stress Disorder Symptoms and the Subsequent Impact of Traumatic Birth on Mother–Infant Relationship Quality

**DOI:** 10.3390/bs14090808

**Published:** 2024-09-12

**Authors:** Lucy J. Frankham, Einar B. Thorsteinsson, Warren Bartik

**Affiliations:** Faculty of Medicine and Health, School of Psychology, University of New England, Armidale, NSW 2350, Australia; lfrankh3@myune.edu.au (L.J.F.); wbartik@une.edu.au (W.B.)

**Keywords:** birth, post-traumatic stress disorder, maternal mental health, personality, childbirth self-efficacy, mother–infant relationship

## Abstract

This study aimed to investigate factors associated with birth-related PTSD symptoms and the subsequent impact on mother–infant relationship quality, exploring what women felt they needed to be different about their birth process to further understand the problem. Participants were recruited using social media advertising. A multi-method approach, using qualitative and quantitative analyses, was employed. The study included 142 pregnant women residing in Australia aged between 19 and 42 years (*M* = 31.24; *SD* = 4.70). High extraversion significantly predicted greater birth-related PTSD symptoms. There was a significant relationship between birth-related PTSD symptoms and poorer mother–infant relationship quality. The content analysis of the responses from women who reported a desire for a different or improved birth experience generated four themes: (1) less intrusive intervention, (2) better communication, (3) access to more supportive intervention, and (4) better post-birth care. The findings highlight the importance of supporting women’s choices during birth by promoting informed and respectful woman-centred care. Postnatally, the screening and assessment of women should go beyond mood screening and include an assessment of the woman’s response to her birth experience.

## 1. Introduction

It is estimated that nearly half of women report their birth experience as being traumatic and that between 3% and 17% are thought to go on to develop clinically significant symptoms of post-traumatic stress disorder (PTSD) related to birth [1,2,3]. Birth trauma can be defined as an experience that involves events or care during labour and birth that cause deep distress or psychological disturbance of an enduring nature, which may or may not include physical injury [4]. PTSD arising from the events of childbirth is classified according to the same criteria outlined in the Diagnostic and Statistical Manual of Mental Disorders (DSM-5-TR) [5]. There are four types of symptoms: intrusive symptoms, avoidance symptoms, hyperarousal and reactivity symptoms, and cognitive symptoms. This may include re-experiencing associated with the birth, distress triggered by reminders of the birth, avoidance and numbing of those reminders, and changes in mood or cognition, such as problems with trust and self-blame [5]. Women who are traumatised by birth often go on to have other difficulties, including problems with infant feeding and sleep, poorer mother–infant bonding, and problems in the couple’s relationship [6,7,8,9]. Furthermore, the infants of mothers who experience birth-related PTSD symptoms may also have disruptions to their psychosocial and neurodevelopment, leading to ongoing problems [10,11].

Existing studies exploring the individual characteristics of those who experience birth-related PTSD symptoms show mixed results. Some studies have shown that poor maternal mental health may be a vulnerability for birth-related PTSD symptoms, while others have indicated aspects such as autonomy and locus of control during birth to be important [2,12]. Extreme pain, loss of control, poor support, fear of childbirth, and maternal mental health have all been identified as potential risk factors associated with birth-related PTSD [2,13,14,15]. While studies have shown that the objective outcomes of birth (e.g., mode of birth) are less important than the subjective aspects (e.g., perceived support) [2,10], the evidence base is still not well established, and there is inconsistency about which individual characteristics are important for prevention purposes. In an effort to narrow these down, Dekel et al. explored risk and vulnerability factors in a systematic review and found five key categories related to birth-related PTSD: (1) negative perception of birth (fear of birth and low internal locus of control); (2) maternal mental health (mostly antenatal depression); (3) trauma history (of any type); (4) delivery mode, and (5) low social support (including family, partner, and staff). Other potentially important individual characteristics, such as personality traits and self-efficacy—which are known to influence trauma responses and aspects of maternal well-being—were not identified in the study and have been poorly explored in relation to birth-related PTSD [16,17].

### 1.1. Childbirth Self-Efficacy

Evidence suggests that mothers with high levels of self-efficacy tend to exhibit greater confidence and are more likely to attribute their success to their own efforts, whereas those with lower self-efficacy give up more readily and tend to attribute their failures to their own actions or abilities [18]. Few studies have explored childbirth self-efficacy in relation to birth-related PTSD. Of those, low childbirth self-efficacy has been shown to be associated with poorer postnatal adjustment [19] and birth-related PTSD [16,20]. This is likely due to mothers with greater self-efficacy being better able to cope with the various subjective elements of childbirth and experiencing a sense of personal mastery and control. A number of other factors have also been shown to be associated with childbirth self-efficacy, including a woman’s experience of pain [21], obstetric-related factors [22], fear of childbirth [23], and post-traumatic stress symptoms [16]. Self-efficacy has also been shown to serve as a protective factor against PTSD more broadly [24]. As such, childbirth self-efficacy may be an important predictor of how birth is experienced.

### 1.2. Personality Traits

Although no definitive model exists to explain the role of personality traits in mental health, research has shown that personality significantly influences well-being and mental health [25,26]. Research shows that personality traits can influence the onset, progression, and manifestation of PTSD symptoms [27]. Specifically, a review by Jakšić et al. [27] found PTSD symptoms are positively associated with neuroticism. Conversely, the review found PTSD symptoms to be negatively associated with extraversion and conscientiousness. Few studies have explored the relationship between personality traits and birth-related PTSD symptoms. Two studies have demonstrated a positive correlation between neuroticism (low emotional stability) and birth-related PTSD [17,28]. One study found optimism to be negatively related to birth-related PTSD symptoms, although this association has only been tested in one study [17]. Research focused on personality and non-birth-related PTSD indicates neuroticism is a vulnerability for the development of PTSD, and the correlational research focused on birth-related PTSD has also demonstrated a significant positive association, suggesting neuroticism may be important. Given the limited studies that specifically examine personality and birth-related PTSD symptoms, it is not clear which other traits may be important.

### 1.3. Mother–Infant Relationship

Positive health and developmental outcomes for children largely depend on the manner in which they are cared for. Problems with the mother–infant relationship can lead to insecure attachment and, therefore, an increased risk of emotional and physical health problems throughout the lifespan of the child [29,30]. In the early stages of infancy, maternal stress has a more immediate impact on emerging stress response systems and infant development overall, in contrast to the effects observed at later stages of development [31], and the quality of attachment can affect how the nervous system is shaped [32]. An early experience of caregiving that is unresponsive, inconsistent, or harsh can lead to adverse health and developmental outcomes throughout the course of life [30]. Traumatic stress during childbirth can influence the nature of the mother–infant relationship. Mothers who experience traumatic birth often express feelings of emotional detachment and have challenges connecting with others, including their own infants [10]. They tend to avoid any triggers or reminders of the traumatic birth experience, which can even extend to actively disconnecting or distancing themselves from their infants [10]. Evidence suggests a significant association between birth-related PTSD symptoms and poorer mother–infant relationships [9,11].

### 1.4. Theories of Birth-Related PTSD

There are two main theories proposed to explain the development of birth-related PTSD. By focusing more on individual factors, Ayers et al. [33] propose that coping and stress-related factors may be involved in the aetiology of birth-related PTSD symptoms and, therefore, are explained by a diathesis-stress model. Consistent with this model, objective complications of childbirth have been found to be less important than a woman’s subjective aspects of the event [2]. Findings of other studies, e.g., [14,15,33], also indicate that while medical interventions and obstetric difficulties (e.g., instrumental birth) are vulnerabilities for birth-related PTSD, medical status seems to be less important than individual subjective factors.

Further, Beck’s [10] theory of traumatic birth aims to explain the consequences of traumatic birth and the broader impact on partners, birth workers, and infants. The theory was developed by combining studies on birth trauma, creating a sophisticated abstract-level middle-range theory. The theory proposes nine axioms to explain the wider consequences of a traumatic birth: (1) PTSD can develop from birth-related trauma; (2) PTSD symptoms can vary in intensity and duration; (3) traumatic birth can have long-term consequences; (4) birth-related trauma can lead to lashing out at clinicians and significant others; (5) mother–infant interactions can be impacted by PTSD symptoms; (6) breastfeeding problems may arise from birth-related trauma; (7) the anniversary may trigger the re-emergence of symptoms; (8) future births increase anxiety; (9) not all births following are healing.

### 1.5. Hypotheses

Building on the research by Ayers [33] and based on the diathesis-stress model, it is hypothesised that (1) known vulnerability factors of antenatal depression, childhood trauma, low childbirth self-efficacy, and low support (maternal social support and couple satisfaction) will predict birth-related PTSD symptoms and (2) that women who experience more birth-related PTSD symptoms will report poorer quality mother–infant relationships. While personality is a known correlate of mental health and well-being, given the limited available research on personality and birth-related PTSD symptoms, all personality traits will be tested as predictors of birth-related PTSD symptoms as part of the first hypothesis. It is expected that low emotional stability (neuroticism) will predict birth-related PTSD symptoms, while the direction of the other personality traits is unclear.

The study also included a qualitative component to explore women’s views on what they consider important regarding their birth experiences and their perceptions of how birth impacts the quality of their relationship with their baby for a more comprehensive understanding of the research questions.

## 2. Materials and Methods

### 2.1. Design

This study was conducted and reported in accordance with the Strengthening the Reporting of Observational Studies in Epidemiology (STROBE) guidelines [34]. The STROBE checklist was used to ensure the transparent and comprehensive reporting of the study design, methods, results, and discussion.

By drawing on the research perspectives of positivism, this study sought to obtain objective empirical data to address the research questions by employing a multi-method design. Given that birth-related PTSD is a complex phenomenon for which understanding is still evolving, qualitative data were incorporated to triangulate the findings. While positivism often focuses on quantitative data, qualitative data can help researchers understand the nuances, meanings, and contextual factors that quantitative data alone might not capture. This supplementary information can provide additional insights and context to quantitative findings and enhance the credibility and reliability of the overall findings. Further, understanding the context is crucial for interpreting quantitative findings, especially in an evolving area of study, ensuring that the results are meaningful in real-world situations. 

As can be seen in Figure 1, the quantitative component utilised a cross-sectional longitudinal design between 28 weeks pregnant (time 1) and six weeks postnatal (time 2), with 10 predictors (i.e., antenatal depression, childbirth self-efficacy, maternal social support, couple satisfaction, childhood trauma and personality [i.e., emotional stability, conscientiousness, intellect, agreeableness, and extraversion]) measured at time 1 with two dependent variables (i.e., birth-related PTSD symptoms and mother–infant relationship) and the co-variate of postnatal depression measured at time 2. The qualitative component asked two questions related to the participant’s birth experience and one about the connection with their baby in relation to their birth. The qualitative questions were asked at time 2.

### 2.2. Participants and Recruitment

A power analysis was conducted using G*Power v3 [34] to determine the number of participants required to conduct a multiple regression analysis for each hypothesis. By using an alpha level of 0.05 and target power of 0.95 with 10 predictors, the number of participants needed to detect a medium effect was 132. Given that the qualitative component is designed to provide a deeper understanding of any relationships identified in the quantitative data, the sample size for the qualitative component was aligned with the quantitative analysis to ensure complementarity. Participants were recruited using social media between September 2020 and April 2021. All advertising adhered to the university's ethical requirements. Participants were asked to complete an online survey about their birth experiences, and an entry into a draw for a gift voucher was offered as an incentive. Informed consent was obtained prior to the survey. Eligibility included being English-speaking, residing in Australia, and being 28 weeks pregnant or greater at the time of taking the first survey. At time 1 (pregnancy), a total of 426 possible participants accessed the survey, 190 attempted the survey, and 142 completed the entire survey, with ages ranging between 19 and 42 years (*M* = 31.24, *SD* = 4.70). Participants were able to skip the childbirth self-efficacy questions if they were having a planned caesarean (*n* = 13). At time 2 (six weeks postnatal), for the 142 participants who completed time 1, 140 attempted the survey, and 130 completed the survey. The participants were asked if they experienced labour and birth to account for those that had either a planned caesarean or an emergency caesarean where no labour and birth was experienced. This decision was based on feedback from an early participant who felt they would have preferred the option not to answer that section of the survey after having had an emergency caesarean. These participants were able to skip the questions on trauma symptoms relating to their birth (*n* = 12).

### 2.3. Materials

#### 2.3.1. Childbirth Self-Efficacy

The abbreviated Childbirth Self-Efficacy Inventory (CBSEI-32) was employed to assess childbirth self-efficacy and expectations, reflecting a woman’s confidence and coping skills during childbirth [35]. The scale consists of two subscales, asking about outcome expectancy (how helpful the behaviour could be) and efficacy expectancy (how certain of their ability to use the behaviour), containing 16 statements each (e.g., ‘Relax my body’). The CBSEI-32 outcome expectancy subscale is scored on a 10-point rating scale from 1 (not at all helpful) to 10 (very helpful) and the efficacy expectancy subscale from 1 (not at all sure) to 10 (very sure). In this study, a composite score was calculated by summing the scores from both scales, yielding a maximum score of 320. Higher scores indicate higher levels of childbirth self-efficacy among pregnant women in relation to labour. The CBSEI-32 has good internal consistency (α = 0.91) [36] and in the present study (α = 0.96).

#### 2.3.2. Personality Traits

Personality was assessed using the International Personality Item Pool (IPIP) Five Factor Markers—50 Item Version, a 50-item questionnaire that measures five dimensions of personality: extraversion, agreeableness, conscientiousness, emotional stability (low neuroticism), and intellect (openness to experience), each with 10 items [37]. Respondents are asked how accurately each of the items describes them (e.g., ‘Like order’), using a 5-point Likert scale ranging from 1 (very inaccurate) to 5 (very accurate); higher scores indicate more of the trait present. Scores for each trait range from 10 to 50. The five IPIP scales have been found to have high internal consistencies: Extraversion (α = 0.87), Agreeableness (α = 0.82), Conscientiousness (α = 0.79), Emotional Stability (α = 0.86), and Intellect (α = 0.84) and in the present study: Extraversion (α = 0.89), Agreeableness (α = 0.78), Conscientiousness (α = 0.74), Emotional Stability (α = 0.75), and Intellect (α = 0.77).

#### 2.3.3. Antenatal and Postnatal Depression

The Edinburgh Postnatal Depression Scale (EPDS) was used to measure depressive symptoms [38]. The EPDS is a 10-item self-report questionnaire; responses are rated on a 4-point Likert scale, ranging either from 0 (no, never) to 3 (yes, most of the time) or from 0 (as much as I ever did) to 3 (hardly at all), and three items are reverse scored (e.g., ‘I have looked forward with enjoyment to things’). The total scores have a range of 0 to 30, where higher scores reflect increased distress levels. The EDPS has been validated for screening depression in pregnancy and postnatally [39,40]. It has been shown to have good psychometric properties [41], including reliability in [40]. Cronbach’s alpha in the present study was 0.89 to 0.90.

#### 2.3.4. Birth-Related PTSD Symptoms

Birth-related PTSD symptoms were measured with the 22-item Impact of Events Scale–Revised (IES-R) [42]. Items (e.g., I tried not to think about it) are rated on a 5-point Likert scale ranging from 0 (not at all) to 4 (extremely), with total scores ranging from 0–88. Higher scores indicate greater trauma symptoms. The IES-R assesses post-traumatic stress disorder symptoms in line with the Diagnostic and Statistical Manual for Mental Disorders—Fourth Edition (DSM-IV) [43]. Although the diagnostic criteria for PTSD have been revised in the DSM-5-TR, including splitting the avoidance and numbing criteria and adding more associated symptoms, these changes do not alter the essential characteristics of PTSD, and the IES-R remains valid for assessing PTSD symptoms [5,44]. Participants were instructed to respond to the questions in relation to their birth. The IES-R demonstrates strong reliability in postnatal women (α = 0.91) [16] and has high internal consistency (α = 0.96) [45]. Cronbach’s alpha for the present study was 0.94.

#### 2.3.5. Maternal Social Support

Social support was measured using the Maternity Social Support Scale (MSSS) [46]. The MSSS contains six questions about social connections with family friends and partners (e.g., I have good friends who support me). Each item is scored on a 5-point Likert scale from 1 (never) to 5 (always), with a possible total score ranging from 6 to 30. Two items are reverse-scored. Higher scores indicate increased support. The MSSS has been found to have good internal consistency (α = 0.76) [47], with adequate reliability in the present study (α = 0.66). The MSSS is a significant predictor of maternal health-related quality of life during pregnancy and following childbirth.

#### 2.3.6. Mother–Infant Relationship

The quality of the mother–infant relationship was measured using the Postpartum Bonding Questionnaire (PBQ) [48]. The PBQ consists of 25 items (e.g., I feel close to my baby). Responses are rated on a 6-point scale ranging from 0 (always) to 5 (never), with total scores ranging from 0 to 125. A total of 17 items are reverse-scored, and higher scores indicate poorer bonding. The PBQ has adequate reliability, good internal consistency (*r* = 0.84 to 0.85), and acceptable construct validity [49,50]. In this study, Cronbach’s alpha was 0.93.

#### 2.3.7. Couple Satisfaction

Satisfaction with the couple’s relationship was assessed using the Revised Dyadic Adjustment Scale (RDAS). The RDAS consists of thirteen items (e.g., ‘Making major decisions’) that are scored on a six-point rating scale ranging from 0 (always disagree) to 5 (always agree), and one item is scored on a five-point scale ranging from 0 to 4 [51]. Higher scores indicate better dyadic adjustment. Total scores range from 0 to 69. The RDAS has been shown to have acceptable levels of construct validity, adequate internal consistency (α = 0.90), and excellent split-half reliability (Guttman split half = 0.94). Cronbach’s alpha for the present study was 0.82.

#### 2.3.8. Childhood Trauma

The Adverse Childhood Experiences (ACE) questionnaire was used to assess childhood trauma. The ACE comprises 10 items, which encompass five items related to various forms of maltreatment (such as physical, sexual, or emotional abuse, as well as physical or emotional neglect) and five items focused on family dysfunction (including parental separation or divorce, exposure to domestic violence, and the incarceration, substance abuse, or mental illness of a household member). These experiences are considered in the context of events occurring between birth and 18 years of age [52]. Respondents answer each question with a yes or no, with the total number that is endorsed as yes summed for a score out of 10. For each category of abuse and family dysfunction, the ACE questionnaire has good test–retest reliability, with kappa coefficients between 0.46 and 0.86 [53]. The total ACE score is generated by adding up all the items that participants have affirmed, with higher scores indicating a higher degree of adverse childhood experiences. Cronbach’s alpha in the present study was 0.76.

#### 2.3.9. Qualitative Questions

Two questions were asked about the women’s childbirth experiences: “Is there anything that could have improved your birth experience?” and “Are there any aspects of your birth that you would prefer to have been different?”. The questions were analysed jointly. One further question was asked about the birth experience in relation to the mother–infant relationship: “Do you feel as though your birth experience influenced the connection you have with your baby? If so, in what way?”, which was analysed separately.

#### 2.3.10. Design

This study employed a mixed-method design. As can be seen in Figure 1, the quantitative component utilised a cross-sectional longitudinal design between 28 weeks pregnant (time 1) and six weeks postnatal (time 2), with 10 predictors (i.e., antenatal depression, childbirth self-efficacy, maternal social support, couple satisfaction, childhood trauma, and personality (i.e., emotional stability, conscientiousness, intellect, agreeableness, extraversion)) measured at time 1 and two dependent variables (i.e., birth-related PTSD symptoms and mother–infant relationship) and the co-variate of postnatal depression measured at time 2. The qualitative component asked two questions related to the participant’s birth experience and one about the connection with their baby in relation to their birth. The qualitative questions were asked at time 2.

#### 2.3.11. Data Analysis

Statistical analyses were performed using IBM SPSS Statistics v27 for Windows [54], and the data were uploaded to Figshare [55]. Assumption testing was carried out on all variables.

Missing data were checked using Missing Value Analysis in SPSS (Version 27.0), and the dataset was found to have no missing data. There were no issues with order dependence (Durbin-Watson values ranged from (1.7 to 2.1). A number of variables were correlated; however, no issues with multicollinearity were detected (Variance Inflation Factor values were all <2.1, and tolerance values were all >10).

Prevalence rates for post-traumatic stress disorder were calculated based on recommended cutoff scores for the Impact of Events Scale-Revised [42]. For multiple regression analysis, if demographic variables did not covary with the dependent variable or independent variables, they were not entered into the analysis.

The qualitative data were analysed using a content analysis approach following Stemler’s method [56] to identify trends and patterns within the three qualitative questions. This method was chosen due to the study’s concise survey design. The data were examined inductively to search for patterns of differences and similarities from which categories were derived. Category development and coding were conducted by the first author and reviewed by all co-authors.

## 3. Results

### 3.1. Participant Characteristics

The sample was comprised predominantly of Caucasian first-time mothers residing in urban Australia. Nearly all the participants were partnered, four were not in a relationship, and three reported being in a same-sex relationship. Whilst this study refers to mothers and women, we recognise that there may be some participants who do not identify as female. Postnatally, 42.3% reported an unassisted vaginal birth, with most women having a medically assisted birth. Less than one quarter described their birth as being consistent with their preferences. Three women did not breastfeed from the beginning, while 16 had stopped by the time of the second survey at six weeks postnatal. The demographic and birthing-related characteristics of the participants are reported in Table 1 and Table 2, respectively.

### 3.2. Preliminary Analysis

A post-hoc power analysis was conducted using G*Power (version 3.0.10). A sample size of 117 with a maximum of 10 predictors was used for the statistical power analyses for the *F*-test for multiple regression. The recommended effect sizes used for this analysis were small (*f*^2^ = 0.02), medium (*f*^2^ = 0.15), and large (*f*^2^ = 0.35). The alpha level used for the current analysis was set at 0.05. The post-hoc analyses revealed the statistical power for this study was 0.80 for detecting a moderate effect, whereas the power exceeded 0.99 for the detection of a large effect size. Thus, there was more than adequate power (i.e., power was >0.80) at the medium-to-large-effect size level. There was inadequate power (0.13) to detect a small effect (*f*^2^ ≤ 0.02).

### 3.3. Prevalence of Post-Traumatic Stress Disorder Symptoms

By applying the recommended cutoff scores on the IES-R (≥33 for probable PTSD and ≥24 for clinically significant symptoms), 10.8% of women indicated IES-R scores that suggested probable PTSD related to their childbirth experiences, while 20.8% reported clinically significant symptoms. The means, standard deviations, and correlations are presented in Table 3.

### 3.4. Predictors of Birth-Related PTSD Symptoms

A multiple regression analysis was conducted to examine whether the independent variables of childbirth self-efficacy, antenatal depression, couple satisfaction, maternal social support, childhood trauma, and personality (i.e., emotional stability, conscientiousness, intellect, agreeableness, extraversion) can predict birth-related PTSD symptoms. As seen in Table 4, the multiple regression model with 10 predictors produced *R*^2^ = 0.09, *F*(10, 116) = 1.02, and *p* = 0.43, and only the personality trait of extraversion significantly predicted birth-related PTSD symptoms (Beta = 0.25; *p* = 0.03), indicating that greater extraversion predicts more birth-related PTSD symptoms. Apart from childbirth self-efficacy, which was found to be in the opposite direction, the remainder of the predictors in the analysis showed trends consistent with the first hypothesis, with each of them found to be in the hypothesised direction.

Additionally, as emotional stability (neuroticism) had been predicted to be significant and was correlated with extraversion, which was a significant predictor, it was decided to run the analysis without each variable to better assess their unique contribution to the model. In the first analysis, with extraversion removed, emotional stability remained non-significant, whereas when emotional stability was removed from the model, extraversion remained significant. The subsequent analyses did not make a substantive change to the interpretation of the original regression analysis; therefore, the results from the original analysis were retained and interpreted.

### 3.5. Birth-Related PTSD Symptoms and the Mother–Infant Relationship

As can be seen in Table 5, the multiple regression model, including the predictor variable of birth-related PTSD symptoms and control variable of postnatal depression (Time 2), produced *R*^2^ = 0.31, *F*(2, 127) = 28.69; *p* < 0.001, indicating that the overall model accounted for nearly one-third of mother–infant relationship quality. Postnatal depression (Beta = 0.45; *p* < 0.001) and birth-related PTSD symptoms (Beta = 0.18; *p* = 0.04) significantly predicted scores for mother–infant relationship quality. Consistent with the second hypothesis, this indicated that higher levels of postnatal depression and birth-related PTSD symptoms predict poorer mother–infant relationship quality; however, due to the small *R*^2^, this result should be interpreted with caution.

### 3.6. Qualitative Findings

The qualitative analysis of responses from women (the first two questions about birth experience) expressing a desire for a different or improved birth experience yielded four key themes, described in Table 6. Some responses were categorised into multiple themes as being relevant. A consistent finding across the dataset was a preference to have reduced obstetric and medical intervention. The responses also highlighted instances where women felt they did not receive sufficient or appropriate intervention when they wanted it, in particular, access to pain relief and water (e.g., birthing pool and shower). Around one-quarter of women desired improved communication, either feeling unheard or as though the information was absent, unclear, or confusing, and 15 women considered post-birth care to be important and sometimes neglected. Those women indicated they wanted better post-birth care.

For the third question, “Do you feel as though your birth experience influenced the connection you have with your baby? If so, in what way?”, most women reported either a positive or negative impact, with less than half providing neutral responses, indicating that women made clear connections between how their birth was experienced and the quality of the bond with their baby (see Table 7). Women who reported that their birth had a positive influence on the relationship with their baby generally described births that were positive, calm, and peaceful. Some of those women spoke about being able to have skin-to-skin time as important, and another spoke of the water being helpful. Conversely, the women who spoke about the connection with their baby being negatively impacted drew associations with feeling traumatised by their birth and not being able to have skin-to-skin time, as well as pain and medication effects. Nine women either did not answer or provided an unclear answer to this question.

## 4. Discussion

The present study investigated predictors of birth-related PTSD symptoms and the subsequent impact on the mother–infant relationship. Themes around what women felt they needed to be different about their births were derived to further understand the problem. Our quantitative findings only partially supported the stress-diathesis model and an explanation for the development of birth-related PTSD symptoms, while the qualitative findings were indicative of other contributing factors.

For the first hypothesis, only extraversion significantly predicted birth-related PTSD symptoms, suggesting that individuals with higher levels of extraversion may be more susceptible to experiencing traumatic birth events. There is little research in relation to extraversion and birth-related PTSD to explain this finding. Extraversion has previously been shown to be negatively related to PTSD symptoms [57] and generally associated with more positive well-being [25]. Further, extraversion has been found to be associated with a lower fear of childbirth prenatally [58]; however, there is no research to suggest that subjective birth experience postnatally differs as a function of extraversion. One study showed extraverted mothers were more likely to have a normal vaginal birth [59]; however, the study did not explore how women with extraverted traits cope in the face of a difficult birth.

One explanation for our finding could be that some aspects of extraversion that are typically viewed as positive, such as sociability and internal locus of control, may indeed be a vulnerability in the face of a difficult birth. Extraverted individuals often value autonomy and control over their environments. In the context of childbirth, where there is a degree of unpredictability and relinquishing of control, extroverts may find it more challenging to adapt to this loss of control and may experience increased stress or anxiety. This difficulty in adjusting to a situation where they have limited control could potentially contribute to perceiving the birth as traumatic. In the context of trauma, extroverts are thought to rely more on their sociability and active engagement to avoid confronting distressing emotions associated with the traumatic event [57]. However, using sociability in this way during birth may be a less effective way of coping with the distress associated with childbirth.

The relationship between childbirth self-efficacy and birth-related PTSD symptoms was predicted to be negatively related; however, it was found to be the opposite of the predicted direction, although the effect size for this finding was small and not statistically significant. This finding is inconsistent with previous research, demonstrating the protective effects of self-efficacy [24]. One explanation for this could be related to the uniqueness of birth-related trauma compared with other trauma (e.g., combat, violence, and accidents). Birth is an everyday event that is typically viewed as positive and where elements of control and safety are anticipated; therefore, factors that are typically protective of PTSD possibly may vary in the context of a traumatic birth. Of relevance to the findings in this study, childbirth self-efficacy has been found to be associated with extraversion [60], which may offer some explanation for the unexpected findings for these two variables in our study. Furthermore, in line with prior studies, e.g., [2,12,14], maternal mental health (antenatal depression and childhood trauma) and social support (maternal social support and couple satisfaction) were found to be weakly associated with birth-related PTSD in the predicted direction, but these relationships were also nonsignificant with small effect sizes.

Consistent with previous research [9], there was a strong relationship between birth-related PTSD symptoms and the mother–infant relationship when controlling for postnatal depression, indicating that the effect of birth-related PTSD symptoms is independent of postnatal depression. This was further supported in the women’s responses about how their birth experience influenced the bond with their babies. In their responses, women were readily able to draw a connection between their birth and how they felt about their baby, and many made that connection in relation to a difficult birth experience and subsequent poorer mother–infant relationship quality. This finding was again independent of depressed mood, which was not mentioned by any of the participants. These results emphasise the importance of screening for birth-related PTSD symptoms in addition to postnatal depression. This is important because the symptoms of birth-related PTSD can impact the mother–infant dyad in unique ways that are different to postnatal depression. For example, responses to a traumatic birth may involve actively avoiding the baby, being triggered by the baby, or feeling too numb to connect [9]. Treatment for trauma is also different. While prevalence rates for birth-related PTSD are contested, even at the lowest reported rates, it warrants giving attention to birth-related PTSD symptoms independently of postnatal depression in order to promote a healthy and positive bond between mothers and their infants.

The remaining qualitative findings aligned with previous research, emphasizing the importance of autonomy and control during labour and birth [2,10,12,61]. The themes of less intrusive intervention, access to more supportive intervention, and better communication reflect the value of autonomy and control in shaping women’s birth experiences. These themes suggest that the women had indeed experienced insufficient autonomy and control over their birth. Less than half of the women reported an unassisted vaginal birth, and less than a quarter reported their birth to be consistent with their preferences, which is also reflected in their responses. The desire for less intervention and better communication resonates with the growing emphasis on promoting individualised woman-centred care and informed decision-making approaches in maternity care. The contrasting preference for more intervention highlights the need for individualised care that respects women’s unique needs and preferences, their right to adequate pain relief, and their capacity to choose. Women clearly wanted more informed care and to be respected in their capacity to choose that care. Finally, the theme of better post-birth care emphasises the significance of comprehensive support during the early postpartum period, encompassing physical and emotional aspects of recovery, bonding, and breastfeeding support. In keeping with the other themes, this highlights that birth is not a medical event; it is a physiological and psychological process that encompasses a range of physical, social, and emotional needs.

For the question that asked about the influence of women’s birth experiences on the connection with their baby, women indicated clear links between their experience and the quality of the bond with their baby. Women who reported a positive connection used words such as ‘peaceful’ and ‘calm’ to describe their births, while women reporting poor quality bonding cited traumatic births without skin-to-skin time or feeling as though pain or medication effects got in the way of bonding. These responses show that women believe what happens during birth makes a difference in how they subsequently bond with their babies.

Both the quantitative and qualitative findings of this study indicate that the stress-diathesis model alone may be insufficient for explaining the development of birth-related PTSD symptoms. Understanding the causes and risk factors for birth-related PTSD symptoms may be more effectively explained by a broader model, such as the Power Threat Meaning Framework (PTMF) [9], which can incorporate explanations from both individual and systemic factors. The PTMF is a comprehensive and empowering framework for comprehending the impact of power imbalances on individual mental well-being resulting from social, cultural, and political elements. Departing from conventional psychiatric paradigms, the PTMF regards psychological distress as a result of what individuals have encountered in terms of power dynamics (power), how it has affected them (threat), the significance they attribute to it (meaning), and their means of surviving or coping with it [62,63]. In the birth space, it could allow for examination of how power imbalances, both within healthcare systems and society, interact with individual characteristics to impact the experience of birth and subsequent psychological well-being and/or development of birth-related PTSD symptoms. Individual factors may relate to existing maternal vulnerabilities and the subjective experience of birth, while systemic issues that influence autonomy and choice may relate to problems such as medical paternalism, oppression of women, and obstetric violence [2,64]. For instance, existing maternal mental health problems, such as anxiety or depression, may influence how individuals perceive and respond to the challenges and uncertainties of childbirth, potentially intensifying their emotional and psychological reactions. Such factors may amplify the emotional impact of the power dynamics inherent in the birthing environment, which can be explained by the PTMF.

### 4.1. Clinical Implications

Being aware that extroverted women may be more vulnerable to the potential negative effects of the birth experience may help clinicians to better guide, assess, and support extroverted women. An important aspect of clinical consideration is being aware that while extraversion is often viewed as a positive and resilient trait, this may not be the case in the context of a difficult birth, and clinicians should be careful not to make this assumption. Considerations such as planning around ways to communicate and creating a sense of control during labour, including managing birth preferences and expectations, as well as sensitive support for coping with medical procedures, may be needed for those with more extroverted traits.

Furthermore, the improvements and changes that women were seeking in their responses highlight how crucial it is for women to have control over interventions and better communication during the birthing process to promote positive birth experiences and outcomes. This is consistent with the World Health Organisation’s recommendations on Intrapartum Care for a Positive Birth Experience [65], which refer to respectful maternity care that involves supporting informed decision-making and responding to women’s preferences. When women feel empowered and involved in decision-making, they are more likely to have a sense of ownership and agency over their birth journey. Having control over intervention options allows women to make informed choices that align with their preferences and values, reducing the risk of unnecessary or unwanted medical interventions and unnecessary pain and discomfort.

Clinicians can play a pivotal role in advocating for a shift away from medical paternalism by acknowledging the rights of women to make choices during the birthing process. Traditionally, medical paternalism has positioned healthcare providers as the sole decision-makers, often prioritizing medical protocols and interventions over the autonomy and preferences of women. By embracing a respectful woman-centred care approach, clinicians can empower women to actively participate in decision-making and honour their rights to make informed choices. Better incorporating woman-centred maternity care can contribute to enhancing women’s experiences and satisfaction with their birth journeys. By addressing these desires for improvement and change, healthcare providers can work towards providing care that aligns with each woman’s preferences, thereby promoting more positive birth experiences.

In regard to the mother–infant relationship, the qualitative and quantitative findings both emphasise the importance of recognizing and addressing birth-related PTSD symptoms as a discrete factor that can affect the early bonding between mothers and their infants. These results highlight the need for healthcare professionals to screen for and address birth-related PTSD symptoms alongside postnatal depression as part of comprehensive care for mothers and infants. Moreover, these findings call for a multidisciplinary approach in maternal and infant healthcare. Collaboration between obstetricians, midwives, psychologists, and other healthcare professionals can help ensure that the impact of birth-related PTSD symptoms on the mother–infant relationship is adequately recognised and addressed. Integrating trauma-informed care practices and providing resources for trauma-focused interventions can further contribute to the overall well-being of both mothers and infants.

### 4.2. Strengths and Limitations

The present study employed a multi-method approach, which allowed for a more comprehensive understanding of factors influencing women’s birth experiences and the subsequent impact on mother–infant relationship quality. The robust, standardised measures are utilised to enhance the validity and reliability of the findings, while the reflexive qualitative approach adds richness and clarity. The combination of pre and post-birth measures, along with the high retention rate, are also important strengths. The study addressed a gap in the literature by including an analysis of childbirth self-efficacy and personality traits, which provides new insight into vulnerability factors that contribute to the development of birth-related PTSD symptoms, specifically in relation to the trait of extraversion, which was found to be a potentially important predictor to explore further.

A limitation of the present study is its use of a cross-sectional design with a relatively small convenience sample recruited through social media. The sample also primarily consisted of university-educated, partnered Caucasian women residing in urban areas, thus decreasing the generalisability of the findings. The overall regression model predicting birth-related PTSD symptoms had low explanatory power and should be interpreted with caution. The independent variables employed accounted for a small proportion of the variance in traumatic birth experiences, suggesting that other factors not included in the model may also play an important role in predicting birth-related PTSD symptoms. Future research using a larger sample and exploring additional variables such as systemic factors (e.g., obstetric violence) is needed to better understand how and under what circumstances women are more likely to experience birth as traumatic.

## 5. Conclusions

This study contributes insights into individual characteristics as predictors of birth-related PTSD symptoms and the subsequent impact on the mother–infant relationship. The findings highlight the importance of promoting autonomy, control, and communication during birth and delivering personalised and woman-centred care that may be better understood by the PTMF. The screening and assessment of women should go beyond mood screening and include an assessment of the woman’s response to her birth experience. By incorporating these recommendations into clinical practice, healthcare professionals can strive to enhance the well-being and satisfaction of women and their infants during the birthing process, thus potentially reducing the chances of future problems.

## Figures and Tables

**Figure 1 behavsci-14-00808-f001:**
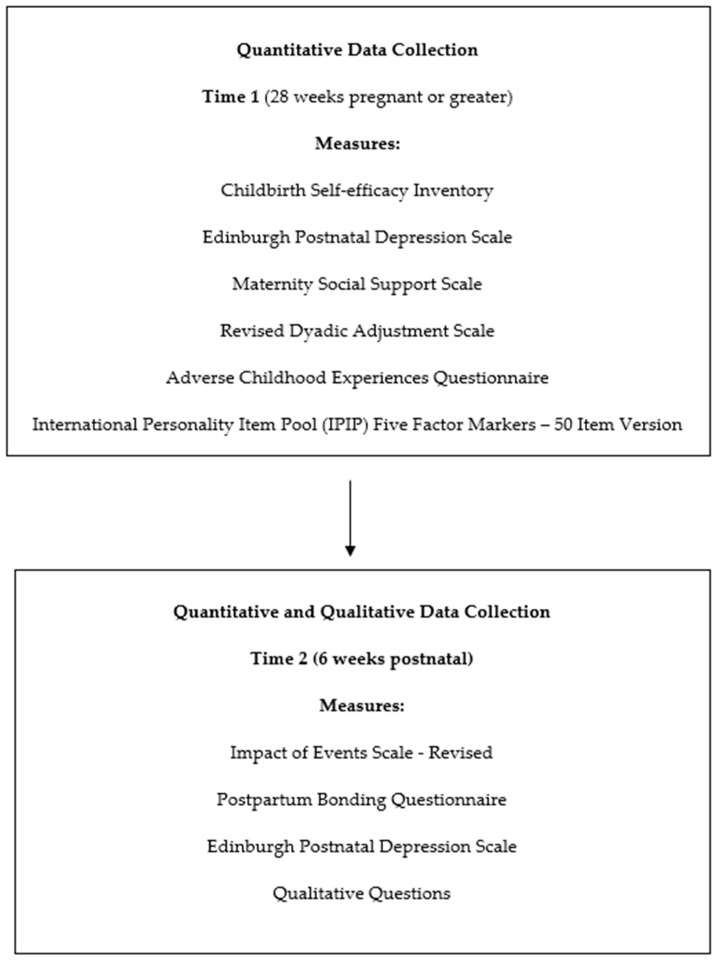
Flowchart of study design and phase.

**Table 1 behavsci-14-00808-t001:** Sample characteristics.

Characteristic	*n*	%
Ethnic background		
Aboriginal or Torres Strait Islander	2	1.4
White European	100	71.4
Indian	2	1.4
Asian	18	12.7
Middle Eastern	1	0.7
South American	1	0.7
Mixed race	9	6.3
Other	9	6.3
Geographical location		
Urban/City	104	74.3
Rural	32	22.9
Remote	6	4.2
In a relationship		
Yes	138	97.2
No	4	2.8
Same-sex relationship	3	2.1
Education		
No formal qualifications	1	0.7
Completed high school	11	7.7
TAFE certificate/diploma	37	26.4
University degree	93	65.5
Total	142	

**Table 2 behavsci-14-00808-t002:** Pregnancy and birthing-related characteristics of participants.

Characteristic	*n*	*%*
Antenatal		
Parity		
Nulliparous	126	88.8
Multiparous	16	11.3
Number of babies		
Singleton	139	97.9
Multiple birth	3	2.1
Birth education classes		
Yes	90	63.4
No	52	36.6
Total	142	
Postnatal		
Type of birth		
Unassisted vaginal birth	55	42.3
Assisted vaginal birth	30	23.1
Caesarean section—emergency	36	28.0
Caesarean section—planned	9	7.00
Birth consistent with preferences		
Yes, completely	24	18.5
Yes, mostly	52	40
No	54	41.5
Breastfeeding		
Yes	111	85.4
Initially, ceased ≤ 6 weeks	16	12.3
No	3	2.3
Total	130	

**Table 3 behavsci-14-00808-t003:** Means, standard deviations, and correlations of childbirth self-efficacy, couple relationship, antenatal and postnatal depression, birth-related PTSD symptoms, mother–infant relationship, and personality traits.

Variable	*M*	*SD*	1	2	3	4	5	6	7	8	9	10	11	12
1. Childbirth Self-efficacy	223.40	46.16	-											
2. Couple Relationship	42.76	2.82	−0.07	-										
3. Maternal Social Support	26.38	2.78	0.17	−0.15	-									
4. Childhood Trauma	1.96	1.96	−0.02	0.01	−0.02	-								
5. Antenatal Depression	7.37	5.36	−0.21 *	0.18 *	−0.41 **	0.14	-							
6. Postnatal Depression	8.39	5.81	0.21	−0.08	0.09	0.02	0.06	-						
7. Birth Trauma	13.90	14.46	0.01	−0.04	−0.10	−0.01	0.16	0.49 **	-					
8. Mother–Infant Relationship	15.93	11.00	0.15	−0.17	0.03	0.11	−0.01	0.54 **	0.40 **	-				
9. Agreeableness	29.59	5.61	0.28 *	−0.06	0.19 *	−0.09	−0.10	0.06	−0.12	−0.06	-			
10. Conscientiousness	26.51	5.93	0.30 **	−0.08	0.19 *	−0.10	−0.29 **	0.02	−0.05	0.12	0.36 **	-		
11. Intellect	27.01	5.72	−0.14	0.11	0.21	0.05	−0.02	0.03	−0.07	−0.09	0.40 **	0.18 *	-	
12. Emotional Stability	19.29	6.73	0.30 **	−0.10	0.32 **	−0.12	−0.61 **	0.04	−0.11	0.01	0.25 **	0.32 **	−0.46	-
13. Extraversion	19.23	7.99	0.04	0.04	0.23 **	−0.18 *	−0.14	0.05	0.92	0.02	0.42 **	0.07	0.32 **	0.34 **

Note. *N* = 117–142; * *p* < 0.05. ** *p* < 0.01.

**Table 4 behavsci-14-00808-t004:** Multiple linear regression analysis predicting birth trauma symptoms.

Predictor	B	Beta	*t*	*SE B*	95% CI for B	*P*
Antenatal Depression	0.36	0.13	1.03	0.34	[−0.33, 1.05]	0.31
Maternal Social Support	−0.44	−0.08	−0.80	0.55	[−1.53, 0.66]	0.43
Childhood Trauma	0.06	0.01	0.82	0.70	[−1.35, 1.46]	0.94
Couple Relationship	−0.51	−0.10	−1.01	0.50	[−1.49, 0.48]	0.31
Childbirth Self-efficacy	0.30	0.09	0.88	0.03	[−0.04, 0.10]	0.38
Extraversion	0.46	0.25	2.22	0.21	[0.05, 0.88]	0.03
Emotional Stability	−0.21	−0.09	−0.70	0.30	[−0.80, 0.38]	0.49
Intellect	−0.20	−0.08	−0.72	0.28	[−0.75, 0.35]	0.48
Conscientiousness	0.13	0.05	0.45	0.28	[−0.42, 0.67]	0.65
Agreeableness	−0.44	−0.17	−1.38	0.32	[−1.07, 0.19]	0.17

**Table 5 behavsci-14-00808-t005:** Multiple linear regression analysis for predicting mother–infant relationship quality.

Predictor	B	Beta	*t*	*SE B*	95% CI for B	*p*
Postnatal Depression	0.85	0.45	5.34	0.16	[0.54, 1.17]	<0.001
Birth Trauma	0.13	0.18	2.08	0.06	[0.01, 0.26]	0.04

**Table 6 behavsci-14-00808-t006:** Themes derived from the questions, “Is there anything that could have improved your birth experience?” and “Are there any aspects of your birth that you would prefer to have been different?” (*n* = 131).

Theme	*n*	%
Less intrusive intervention (e.g., induction; forceps)	42	32
Better communication (e.g., being informed; being listened to)	33	25
Access to more supportive intervention (e.g., pain relief; water)	24	18
Better post birth care (e.g., more skin-to-skin time; breastfeeding support; opportunity to debrief)	15	11

**Table 7 behavsci-14-00808-t007:** Themes derived from the question “Do you feel as though your birth experience influenced the connection you have with your baby? If so, in what way?” (*n* = 131).

Theme	*n*	%
No influence or neutral influence	57	44
Birth experience had a positive influence on connection	38	29
Birth experience had a negative influence on connection	27	21
No response or unclear	9	7

## Data Availability

The dataset supporting the conclusions of this article is available in the Figshare repository: https://figshare.com/articles/dataset/Predictors_of_Birth_Trauma/24030648, accessed on 25 August 2023.

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
