# Peer review of "Factors Associated with Birth-Related Post-Traumatic Stress Disorder Symptoms and the Subsequent Impact of Traumatic Birth on Mother–Infant Relationship Quality"

_behavsci, 2024, doi:10.3390/bs14090808_

Round 1

Reviewer 1 Report

Comments and Suggestions for Authors

1.Authors have studied an important issue and need to be congratulated

2. The study is based on 'online' questionnaire. How did the authors check the reliability of feedback from study participants?

3.First assessment was done at 28 weeks and second during postpartum. How many particiapnts could contribute during both periods of assessment?

3.The term ''mixed method' is more suited than multimedia method. There is only one media used in this study

4.Usually there are no side headings in the introduction and discussion. First letter of 'Key words' to be capital. It should be 'Material and methods' and not 'Materials'.

5.Introduction should be brief and specific. Hypothesis also should be brief.

6..Authors have used multiple scales.  Usually it is not easy to get response from participants  for many questionnaires at one time. How did the authors mange the compliance?

7.Authors may put a 'PICO' chart for the study flow

8.Number of Tables can be reduced. Table.1 and 2 have multiple subdivisions but some have only single digit participants. This is likely to impact the analysis and study outcome

9.Manuscript requires major revision to make it  easy for readers to understand the findings

Comments on the Quality of English Language

Requires language editing in some places

Author Response

Thank you for taking the time to review our paper. Please find below our responses based on your feedback.

Comment 1. Authors have studied an important issue and need to be congratulated

Response: We thank you for your kind comment.

Comment 2.  The study is based on 'online' questionnaire. How did the authors check the reliability of feedback from study participants?

Response: Only participants who completed the entire survey and provided a valid email address for the follow up were included. Online recruitment has been included as a limitation in the limitations section.

Comment 3. First assessment was done at 28 weeks and second during postpartum. How many participants could contribute during both periods of assessment?

Response: As stated in the participants and recruitment section “At time one (pregnancy) a total of 426 possible participants accessed the survey, 190 attempted the survey and 142 completed the entire survey”.

Comment 4. The term ''mixed method' is more suited than multimedia method. There is only one media used in this study

Response: Thank you. We have amended this.

Comment 5. Usually there are no side headings in the introduction and discussion. First letter of 'Key words' to be capital. It should be 'Material and methods' and not 'Materials'.

Response: This has been amended accordingly.

Comment 6. Introduction should be brief and specific. Hypothesis also should be brief.

Response: The introduction and hypotheses were reviewed carefully. Given the large number variables, it was decided necessary to include the detail in the introduction and hypotheses to keep it clear.

Comment 7. Authors have used multiple scales.  Usually it is not easy to get response from participants  for many questionnaires at one time. How did the authors mange the compliance?

Response: Thank you. As indicated in the participants and recruitment section, an inducement was offered in return for participation.

Comment 8. Authors may put a 'PICO' chart for the study flow

Response: Thank you. It was decided that since we did not evaluate an intervention a PICO chart was not suitable for our study.

Comment 9. Number of Tables can be reduced. Table.1 and 2 have multiple subdivisions but some have only single digit participants. This is likely to impact the analysis and study outcome

Response: We reviewed the tables. We consider the demographic and birthing related characteristics important for highlighting limitations about generalisability and birthing trends. They contain important detail that should not be removed.

Comment 10. Manuscript requires major revision to make it  easy for readers to understand the findings

Response: Thank you for your comments. We have addressed your concerns as much as possible.

Reviewer 2 Report

Comments and Suggestions for Authors

Author Response

Thank you for taking the time to review our paper. Please find below our responses based on your feedback.

Comment 1. It is suggested that the title be fine-tuned, and I think it might be more appropriate to use "Factors Associated with" or "Contributors to" instead of "Predictors". I think it may be more appropriate to use "Factors Associated with" or "Contributors to" instead of "Predictors", so as to reduce the predetermined judgment on causality.

Response: Agree. Based on this suggestion we have changed it to “Factors associated with…”.

Comment 2. The paper needs to distinguish between independent variables and dependent variables more clearly in the introduction and hypothesis formulation. In particular, the role of birth related PTSD symptoms as a key variable in the study needs to be clearly defined. If birth related PTSD symptoms are considered as a dependent variable, it is recommended that the relationship between the time point of its measurement and the independent variable be explicitly stated in the Methods section and its association with the independent variable be reported in detail in the Results section. If birth related PTSD symptoms are considered as a mediating variable, it is recommended that a mediation analysis framework be used and that the process and results of testing for mediating effects be reported in detail in the methods and results sections.

Response: The authors reviewed the introduction and hypotheses. It was agreed that the dependent variable is clear in the title and hypotheses outlined in the paper and as such no further changes were made. The independent and dependent variables are also clearly stated in the design section, using these explicit terms. Time points for measurement of all variables are outlined in Figure 1, and referred to in the design and recruitment sections of the method.

Comment 3. It is recommended that an explanation be added to the methodology section as to why mothers who plan to have a caesarean section do not have to complete the Childbirth Self-efficacy Inventory. Considering that pregnant women in different cultures may choose to have a cesarean section due to personal preference rather than solely for medical reasons, there may be a correlation between this decision and childbirth self-efficacy. Therefore, an explanation of this choice would help to accommodate readers from different countries and enhance the generalizability of the study.

Response: This was checked and reviewed. The relevant information is included in the last three sentences of the participants and recruitment section. “Participants were asked if they experienced labour and birth to account for those that had either a planned caesarean or an emergency caesarean where no labour and birth was experienced. This decision was based on feedback from an early participant who felt they would have preferred the option not to answer that section of the survey after having had an emergency caesarean. These participants were able to skip the questions on trauma symptoms relating to their birth (n = 12).”.

Comment 4. I noticed an inconsistency between the description of Figure 1 and 2.2.10. in the thesis, which may cause confusion for the reader. It is recommended that the diagrams and textual descriptions be checked and corrected as necessary

Response: Thank you. To avoid repetition, Figure 1 details the scales used to measure each variable, while the text in the design section outlines the variables. Each scale and what it measures is also outlined in the method section.  

Round 2

Reviewer 1 Report

Comments and Suggestions for Authors

1.Authors have answered the questions raised without significant changes in the manuscript. 

2.Multi-media method is still used in the abstract. Does the journal use lower case letters for the first letter of 'Key words'

3.Introduction and hypothesis should be more specific and brief

4.Can the authors combine the variables with small numbers?

5.Strength and limitations of the study can be made more brief and specific

Author Response

Thank you for taking the time to review our paper. Please find below our responses based on your feedback.

Comment 1: Authors have answered the questions raised without significant changes in the manuscript. 

Response: Several changes were made where appropriate and responses given with reasons where changes were not considered appropriate. Further changes have now been made in accordance with your review.

Comment 2: Multi-media method is still used in the abstract. Does the journal use lower case letters for the first letter of 'Key words'

Response: The term ‘multimethod’ is used in the paper and was changed to ‘mixed method’ at last review. The original manuscript never used the term ‘multi-media’ and this term is also not used in the abstract of the current manuscript. The abstract uses the term multimethod. We have changed it to say mixed methods for consistency.

Yes, the journal article template uses lower case for keywords, therefore this has been changed back to lowercase, as per the original manuscript.

Comment 3: Introduction and hypothesis should be more specific and brief.

Response: The introduction and hypotheses were reviewed carefully. Given the large number variables (6 IV’s and 2 DV’s), it was decided necessary to include the detail in the introduction and hypotheses to keep it clear. Removing information is likely to lead to the study aims being unclear.

Comment 4: Can the authors combine the variables with small numbers?

Response: The N for each variable in the analyses range from 117-142. It would not be appropriate to combine any of these variables, especially since they are measuring different constructs with different scales.

5.Strength and limitations of the study can be made more brief and specific

Response: We have removed information from this section to make it briefer.

Round 3

Reviewer 1 Report

Comments and Suggestions for Authors

1.Generally, there are no side headings in the introduction. It should contain available infromation in the research area and the reason for conducting the present study. Some of the information from introduction can be shifted to discussion.

2.Hypothesis should contain what the authors are  expecting from the study and not general statements. The first sentence under hypothesis subheading appears to be hypothesis.

3.Tables:1 and 2 can be combined since they contain general characteristics of the study population.

4.There were three couples with same sex. How the postpartum issues are related to them?. 

Manuscript can be edited 

Author Response

Comment 1: Generally, there are no side headings in the introduction. It should contain available information in the research area and the reason for conducting the present study. Some of the information from introduction can be shifted to discussion.

Response: Sub-headings are commonly used in the Behavioral Sciences Journal and are an important way to improve readability of the paper.

New information generally should not be included in a discussion that has not been introduced in the introduction and therefore cannot be re-ordered.

Comment 2: Hypothesis should contain what the authors are expecting from the study and not general statements. The first sentence under hypothesis subheading appears to be hypothesis.

Response: The hypotheses are written clearly: “(1) known vulnerability factors of antenatal depression, childhood trauma, low childbirth self-efficacy and low support (maternal social support and couple satisfaction) will predict birth related PTSD symptoms and (2) that women who experience more birth related PTSD symptoms will report poorer quality mother-infant relationships.”. This is clear and succinct.

Given the limited available research available about personality and birth related PTSD, the additional paragraph after the hypotheses provides more detailed information for the rationale around including personality in hypothesis one. It is important for hypotheses to be theory driven and this is why we have included this information.

Given there are a total of twelve variables (ten predictors and two dependent variables) contained in the hypotheses, we believe three sentences in total is succinct. We are concerned removing information would lead to the hypotheses not being clear.

Comment 3: Tables:1 and 2 can be combined since they contain general characteristics of the study population.

Response: Tables 1 and 2 display different information and are also far too long to combine. Table 1 includes demographics while Table 2 contains birthing characteristics. Combining these two tables would make the readability of the tables poor.

Comment 4: There were three couples with same sex. How the postpartum issues are related to them?

Response: Same sex couples experience the same postpartum issues as heterosexual couples.